# Nitrogen Dense Distributions of Imidazole Grafted Dipyridyl Polybenzimidazole for a High Temperature Proton Exchange Membrane

**DOI:** 10.3390/polym14132621

**Published:** 2022-06-28

**Authors:** Qi Pei, Jianfa Liu, Hongchao Wu, Wenwen Wang, Jiaqi Ji, Keda Li, Chenliang Gong, Lei Wang

**Affiliations:** 1State Key Laboratory of Applied Organic Chemistry, Key Laboratory of Special Function Materials and Structure Design of Ministry of Education, College of Chemistry and Chemical Engineering, Lanzhou University, Lanzhou 730000, China; peiq20@lzu.edu.cn (Q.P.); 1910342021@email.szu.edu.cn (J.L.); wuhch19@lzu.edu.cn (H.W.); wangww20@lzu.edu.cn (W.W.); jijq21@lzu.edu.cn (J.J.); likd21@lzu.edu.cn (K.L.); 2Shenzhen Key Laboratory of Polymer Science and Technology, College of Materials Science and Engineering, Shenzhen University, Shenzhen 518060, China

**Keywords:** polybenzimidazole, bipyridine, ethyl imidazole, high temperature proton exchange membrane

## Abstract

The introduction of basic groups in the polybenzimidazole (PBI) main chain or side chain with low phosphoric acid doping is an effective way to avoid the trade-off between proton conductivity and mechanical strength for high temperature proton exchange membrane (HT-PEM). In this study, the ethyl imidazole is grafted on the side chain of the PBI containing bipyridine in the main chain and blended with poly(2,2′-[p-oxydiphenylene]-5,5′-benzimidazole) (OPBI) to obtain a series of PBI composite membranes for HT-PEMs. The effects of the introduction of bipyridine in the main chain and the ethyl imidazole in the side chain on proton transport are investigated. The result suggests that the introduction of the imidazole and bipyridine group can effectively improve the comprehensive properties as HT-PEM. The highest of proton conductivity of the obtained membranes under saturated phosphoric acid (PA) doping can be up to 0.105 S cm^−1^ at 160 °C and the maximum output power density is 836 mW cm^−2^ at 160 °C, which is 2.3 times that of the OPBI membrane. Importantly, even at low acid doping content (~178%), the tensile strength of the membrane is 22.2 MPa, which is nearly 2 times that of the OPBI membrane, the proton conductivity of the membrane achieves 0.054 S cm^−1^ at 160 °C, which is 2.3 times that of the OPBI membrane, and the maximum output power density of a single cell is 540 mW cm^−2^ at 160 °C, which is 1.5 times that of the OPBI membrane. The results suggest that the introduction of a large number of nitrogen-containing sites in the main chain and side chain is an efficient way to improve the proton conductivity, even at a low PA doping level.

## 1. Introduction

The proton exchange membrane fuel cell (PEMFC) has been considered one of the most competitive power conversion devices because of its environmental friendliness, high efficiency, mild operating conditions, and wide applications [1,2,3]. As a key component of PEMFC, the proton exchange membrane (PEM) plays a role in blocking fuel and transferring protons to ensure the completion of a redox reaction. Therefore, the high efficiency and stability of PEM are the critical factors that determine the performance of PEMFC [4]. However, Nafion, the state-of-the-art PEM, can only be used under 80 °C, which is prone to cause the CO poisoning of the catalyst. Developing high temperature (120–200 °C) PEMFC (HT-PEMFC) can dramatically reduce the cost and increase the operation temperature of the PEM fuel cell under low humidity conditions [5]. The HT-PEMFC can not only increase tolerance to the CO of the catalyst, but can also simplify the water management system. Phosphoric acid (PA) doped polybenzimidazole (PBI) has been regarded as an attractive candidate for high temperature PEM (HT-PEM) due to its high thermal stability, good mechanical strength, and high proton conductivity, even without humidification [6,7,8]. In PA-doped PBI, the transferring of the proton obeys the “hopping mechanism” and the “vehicle mechanism”. Under anhydrous and high temperature conditions, the proton transport mainly completes by the “hopping mechanism”: hopping from N-H sites of imidazole to PA anions, hopping from one PA anion to another, and hopping along the imidazole of PBI chains [9]. Therefore, the higher the PA-doped level (ADL) in the PBI membrane, the more hopping sites for proton transport, thereby improving the proton conductivity and efficiency of the PEMFC. However, the higher ADL will also weaken the interaction of the PBI chains, lead to excessive swelling and decrease or even lose the mechanical strength of the PEM [10].

In order to avoid the trade-off between conductivity and mechanical strength, many efforts have been made, such as crosslinking [11,12], nano filler composition [13,14,15], and basic groups’ introduction in the PBI main chain or side chain [16,17,18,19]. During these methods, the introduction of basic groups such as pyridine, imidazole, and triazole in the PBI can increase the number of proton hopping sites along the PBI main chain; therefore, the proton conductivity can reach a high level even when the PA ADL is low, and the low ADL can maintain a good strength of the PBI membrane without excessive swelling. Benicewicz et al. [20] incorporated pyridine groups in the PBI main chain, and the resulting PA-doped membranes showed high mechanical strength and proton conductivity. Berber et al. introduced the bipyridine units in the PBI structure and found that the elongation at break and proton conductivity of the membranes were enhanced simultaneously [21]. Recently, Tang et al. prepared a series of grafted PBI containing a benzimidazole side pendant through N-substitution reaction. The results showed that the grafted PBI membranes with side chains of benzimidazole groups exhibited excellent performance in terms of phosphoric acid doping level, phosphoric acid retention and proton conductivity, anti-oxidative stability, and thermal stability. In addition, the molecular dynamics simulation demonstrated that the improved proton conductivity of the grafted membrane was mainly due to the dense structure of the hydrogen bond network between the polymer membrane and phosphoric acid. Moreover, the greater the number of grafted nitrogen-containing groups, the denser the constructed hydrogen bond network structure, which is beneficial for the PA-doped PBI membrane to obtain excellent proton conductivity at low PA [22].

Based on the observations above, this study aims to fabricate a PBI membrane with low PA doping but high proton conductivity. Firstly, the bipyridine group was introduced in the main chain of PBI for increasing the density of nitrogen sites on the backbone, thereby facilitating the construction of long-range ordered proton transport channels. Secondly, in order to further improve the proton conduction of the blend membranes at low PA doping, 1-(2-chloroethyl)-1H-imidazole was grafted on the side chain of the dipyridyl PBI in the hopes of forming a long-range three-dimensional nitrogen site distribution and a denser proton transport network, thus improving the proton conductivity under low PA doping.

## 2. Experimental Section

### 2.1. Synthesis of Dipyridyl PBI (DPPBI) Copolymers

The DPPBI was synthesized from 2,2′-bipyridine-4,4′-dicarboxylic acid (BPDCA), 4,4′-oxybisphthalic acid (OBBA), and 3,3′-diaminobenzidine (DAB) through condensation polymerization, and the mole ratio of OBBA and BPDCA was 1:1. The specific steps of the DPPBI synthesis are as follows: 30 g of polyphosphoric acid (PPA) was added to a 100 mL three-necked flask and mechanically stirred under a nitrogen atmosphere for 1 h until the PPA was homogeneous, transparent, and free of bubbles. In the second step, 6 mmol of DAB monomer was added and the mixture was stirred at 140 °C for 2 h until the DAB was uniformly dispersed. In the third step, 3 mmol of BPDCA and 3 mmol of OBBA monomer were added. After the monomer was stirred and dispersed uniformly, it was heated to 180 °C and mechanically stirred for 48 h. The viscous solution was poured into 500 mL of deionized water, NaHCO_3_ was added to remove the remaining PPA, and then the DPPBI fiber was filtered and thoroughly washed with deionized water.

### 2.2. Grafting the 1-(2-chloroethyl)-1H-imidazole to DPPBI (ImDPPBI-x)

The procedure for grafting the 1-(2-chloroethyl)-1H-imidazole to DPPBI was according to published literature [23]. First, 1.0 g of DPPBI was added to a 25 mL two-necked flask followed by 5 mL of DMAc, heated to 60 °C, and stirred for 2 h until the polymer was dissolved. Then, a certain amount (5%, 10%, and 20% of the polymer mass) of 1-(2-chloroethyl)-1H-imidazole hydrochloride was added. After stirring and reacting for 12 h, the reactant was dropped into ethanol, with a rubber tip dropper to precipitate, and washed several times. The product was dried in a vacuum oven at 120 °C for 12 h. The prepared polymers are named ImDPPBI-5, ImDPPBI-10, and ImDPPBI-20, respectively.

### 2.3. Preparation of the PBI Composite Membranes

To begin, 0.175 g of (2,2-(p-oxydiphenylene)-5,5-bibenzimidazole) (OPBI) powder, 0.175 g ImDPPBI, or DPPBI and 10 mL of DMAc were added to a 25 mL three-necked flask, the temperature controlled at 80 °C, and were stirred magnetically for 3 h until the polymer dissolved uniformly. The solution was poured into a centrifuge tube and a centrifuge was used to remove insoluble matter, then the solution was dropped on a glass plate, dried at 100 °C for 12 h, and peeled off. The prepared membranes, OPBI/DPPBI, OPBI/ImDPPBI-5, OPBI/ImDPPBI-10, and OPBI/ImDPPBI-20, are named COPBI-0, COPBI-5, COPBI-10, and COPBI-20, respectively.

## 3. Results and Discussion

### 3.1. Characterization of Imidazole Grafted Dipyridyl PBI (ImDPPBI-x)

The materials and reagents and characterization methods used in this work are illustrated in the Appendix A. The procedure for the synthesis of imidazole grafted dipyridyl PBI (ImDPPBI-x), where the x is the amount of added 1-(2-chloroethyl)-1H-imidazole hydrochloride in dipyridyl PBI (DPPBI), is shown in Figure 1. The DPPBI precursor was synthesized from BPDCA, OBBA acid monomers, and DAB amino monomer through condensation polymerization. The mole ratio of OBBA and BPDCA was controlled as 1:1 because too much pyridine pendant will cause excessive PA absorption, thereby decreasing the mechanical strength of the membrane. The structures of the DPPBI, ImDPPBI-5, ImDPPBI-10, and ImDPPBI-20 were verified by ^1^H NMR, as shown in Figure 2. For the ungrafted polymer DPPBI, the proton signals of imidazole at 13.5 ppm and 13 ppm are assigned to the H12 of OBBA units and H13 of BPDCA units, respectively. The integration ratio of the H12 to H13 is almost 1:1, which is in good agreement with the designed monomer contents. In order to investigate the effect of different grafting ratios of imidazole on the properties of the prepared PBI membrane as PEM, 5 wt%, 10 wt%, and 20 wt% of 1-(2-chloroethyl)-1H-imidazole hydrochloride to the weight of DPPBI polymer were used in the grafting reaction. The ^1^H NMR of ImDPPBI-5, ImDPPBI-10, and ImDPPBI-20 obviously show that the peaks of protons in imidazole weakened with the increase in 1-(2-chloroethyl)-1H-imidazole hydrochloride added. This indicates that the hydrogen at the -NH of imidazole units was successfully substituted, and the remaining -NH gradually decreased with the increase in the grafting degree. Moreover, the increase in the CH_2_ proton resonance signals of ethyl imidazole at 1.49 ppm and 3.30 ppm also provide evidence that the 1-(2-chloroethyl)-1H-imidazole was successfully grafted on the DPPBI.

To further prove the successful synthesis of grafted DPPBI, all samples were characterized by Fourier transform infrared (FT-IR), and the results are shown in Figure 3. The peaks at 2925 cm^−1^ and 2850 cm^−1^ are assigned to the stretching vibration of methylene groups. It can be seen that DPPBI does not have obvious methylene peaks, and the methylene peaks of ImDPPBI-5, ImDPPBI-10, and ImDPPBI-20 are gradually enhanced, which is consistent with the ^1^H NMR information. The intensity of the peak at 1600 cm^−1^ due to the stretching vibration of C=N on benzimidazole and imidazole, and the peak at 1290 cm^−1^ due to the stretching vibration of C-N, increase gradually with the increase in grafting degree. The above results indicate that DPPBIs with different grafting degrees were successfully synthesized.

### 3.2. Preparation of the PBI Composite Membranes

As mentioned in the published literature, [24] we also found too much pyridine pendant will cause the excessive PA absorption, thereby decreasing the mechanical strength of the membrane or causing the PA leaking during the fuel cell operation. Therefore, the DPPBI, ImDPPBI-5, ImDPPBI-10 and ImDPPBI-20 were blended with OPBI to adjust the comprehensive properties as HT-PEM. The content of OPBI was determined as 50% and named as COPBI-0, COPBI-5, COPBI-10 and COPBI-20, respectively.

### 3.3. The Membranes Morphology and Thermal Stability

The microscopic morphologies of the surfaces and cross-sections of all membranes are shown in Figure 4. The surface morphologies of all membranes were uniform and dense, which indicates that the grafted DPPBI and OPBI still have good compatibility.

The thermal stability of the polymer film is one of the key performance indicators for the long-term stable operation of the fuel cell, and the thermogravimetric analyses of all four types of films and the commercial available OPBI were carried out as shown in Figure 5. All of the COPBI membranes showed higher stability than that of OPBI, and their decomposition temperatures were above 250 °C. Normally, the decomposition temperature would decrease with the increase in the grafted ethyl imidazole because of the un-thermal stability of the aliphatic units. Interestingly, the first stage of the decomposition temperature around 250–300 °C was in the order of COPBI-5 > COPBI-20 > COPBI-10 > COPBI-0 > OPBI. This abnormal phenomenon of the first decomposition temperature can be explained by the strong interaction between the N of the ethyl imidazole and the H-N of the imidazole in the main chain. However, after the temperature increased above 550 °C, the residual mass from COPBI-0 to COPBI-20% gradually decreased. The higher the grafted imidazole group content, the more weight loss, which also proves the gradual increase in the grafting degree from COPBI-5 to COPBI-20.

### 3.4. The Acid Doping Level, Swelling Ratio, and Proton Conductivity

The acid doping content (ADC) and swelling ratio of all polymer membranes are shown in Figure 6, and the specific values are shown in Table 1. It can be seen that the ADC of the COPBI membrane was higher than that of OPBI due to the dense distribution of the basic nitrogen in the COPBI structure. Moreover, the ADC of COPBI membranes gradually increases with the increase in the grafting degree, from 261% increased to 313%, because the small molecules branched on the side chain can expand the space between polymer chains, increase the free volume, and improve the absorption of PA [25]. Meanwhile, the imidazole group grafted on the side chain is a basic group, which can enhance the affinity of the polymer chain with PA and further increase the absorption of PA [22]. As a result, the swelling ratio of the acid-doped film increased from 193% to 246%, which was higher than 174% of OPBI.

Proton conductivity is the key performance of PEMs, and test temperature and ADL are the core influencing factors of proton conductivity. As shown in Figure 7, all of the COPBI membranes exhibited higher proton conductivities than that of the state-of-the-art OPBI. With the increase in temperature, the conductivity of the membranes gradually increased. However, above 160 °C, the conductivity of some membranes decreased, which may be attributed to the dehydration of the PA molecules after 169 °C [26,27]. With the increase in grafting degree, the conductivity of the COPBI membrane gradually increased, and the highest proton conductivities of COPBI-0, COPBI-5, COPBI-10, and COPBI-20 were 0.077, 0.092, 0.101, and 0.105 S cm^−1^, respectively. There are two main reasons for the increase in conductivity: (1) With the increase in grafting degree, the ethyl imidazole side chain increases the free volume of the polymer chains, which allows more PA doping for proton transfer; (2) the basic imidazole group branched on the side chain can increase the density of the hydrogen bond network in the membrane after absorbing PA, thereby increasing the proton transport paths and improving the proton transport efficiency [22,25].

In order to prove that the high conductivity of the COBPPBI-20% membrane can be achieved at low ADL, the OPBI, COPBI-0, and COPBI-20 were immersed in 85% PA at 80 °C for 90 min to investigate the absorption behavior of the membranes. The PA doping contents (ADC) curves of the OPBI, COPBI-0, and COPBI-20 (Figure 8a) shows that COPBI-0 and COPBI-20 exhibit much higher ADC than that of OPBI, which provides evidence that the dense distribution of basic nitrogen in the PBI structure can effectively increase the interaction between PA and PBI chains. Moreover, the absorption rate of PA of COPBI-20% is slower than that of the COPBI-0 membrane, which may be attributed to the interaction between ethyl imidazole and the N-H of the imidazole in the main chain slowing down the PA absorption of the grafted COPBI-20. However, after 60 min PA doping, the COPBI-20 exhibited higher ADC than that of OPBI and COPBI-0, which also gives evidence that the introduction of basic groups is beneficial to achieve the higher ADL. Furthermore, the OPBI, COPBI-0, and COPBI-20 membranes with low ADC (controlled at 177–179%) were obtained by the above PA doping curve of the membranes. As shown in Figure 8b, both the COPBI-0 and COPBI-20 exhibited higher proton conductivities than did OPBI without bipyridine at the same ADC; since the introduction of basic bipyridine groups in the PBI can increase the number of proton hopping sites along the PBI main chain, the proton conductivity can reach a high level even when the PA ADC is low. In addition, in comparison with the conductivity of the OPBI membrane (178% of ADC) and COPBI-0 membrane (177% of ADC), the highest proton conductivity of the ethyl imidazole grafted COPBI-20 (179% of ADC) increased to 0.054 S cm^−1^ at low ADL, which was 2.3 times that of the OPBI membrane. These results demonstrate that the introduction of dense, long-range stereoscopic nitrogen-containing sites within the PBI membrane can effectively improve the conductivity of PA-doped PBI membranes at low ADL. The dense, long-range steric nitrogen-containing sites can greatly increase the density of the intramembrane hydrogen bond network, provide more continuous proton transport channels, and improve the proton conduction efficiency [22].

### 3.5. Mechanical Properties of the Membranes

The mechanical properties of the undoped PA PBI membranes and the PA-doped PBI membranes were evaluated as shown in Figure 9 and Table 1. Each undoped PA PBI membrane exhibited excellent mechanical strength (Figure 9a). Due to the interaction between basic groups of COPBI and NH of imidazole, the COPBI-0, COPBI-5, and COPBI-10 displayed higher tensile strength than that of OPBI. However, with the increase in the degree of grafting, the tensile strength of the membranes gradually decreased and the COPBI-20 exhibited the lowest strength. This result suggests that the side chain-branched small molecules increase the distance and weaken the interaction between polymer chains. After a saturated doping with PA, the tensile strength of all membranes decreased significantly and the elongation at break increased greatly, because of the PA plasticization effect and weakness of the hydrogen bonding between polymer chains (Figure 9b) [11]. The more PA that was doped, the more the tensile strength decreases, but in general, the tensile strength of all PA-doped membranes can meet the requirements of HT-PEMFC operation. In order to prove that the high mechanical strength of the COBPPBI-20% membrane can be achieved at low ADL, the tensile strength and elongation at break of COBPPBI-20% with 179% ADC membrane were tested as shown in Figure 9b. Surprisingly, the tensile strength of the membrane was 22.2 MPa, which was nearly two times that of OPBI with the same ADC.

### 3.6. Fuel Cell Performance

The single-cell performances of the saturated PA-doped OPBI, saturated PA-doped COPBI-0 and COPBI-20 membranes, and low PA-doped COPBI-20 membrane were evaluated at 160 °C under H_2_/O_2_ without humidification. The current-potential and power density curves are shown in Figure 10. As expected, the saturated PA-doped COPBI-20 achieved a maximum power density of 836 mW cm^−2^ at 313% ADC, which is 2.3 times that of the OPBI membrane (358 mW cm^−2^) and 1.3 times that of the ungrafted COPBI-0 membrane (622 mW cm^−2^). The results indicate that the introduction of nitrogen-containing sites in the main chain and side chain is beneficial to the improvement in the performance of PA-doped membranes as HT-PEM. Notably, the cell performance under low PA doping was also tested, and the maximum output power of the single cell reached 540 mW cm^−2^, which was 1.5 times that of the OPBI membrane. The result suggests that even at the same low PA doping, there are more continuous and dense proton transport channels in the COPBI-20 membrane, which facilitate the good performance in HT-PEMFC operation.

The comprehensive properties of the PA-doped COPBI-0 and COPBI-20, including the tensile strength, ADC, proton conductivity, and the highest power density, were compared with those of commercially available OPBI and other published literature related to pyridine-containing or -grafting PBI membranes (Table 2). The results illustrated that the COPBI-20 with low ADC (179%) successfully avoids the trade-off between conductivity, power density, and tensile strength. While the tensile strength of the low acid-doped COPBI-20 was maintained at 22.2 MPa, the highest power density can still be as high as 540 mW cm^−2^ at 160 °C.

## 4. Conclusions

The ethyl imidazole was grafted on the side chain of the dipyridyl PBI and blended with OPBI to obtain a series of polymer composite membranes for HT-PEMs. The effects of the introduction of bipyridine in the main chain and the ethyl imidazole in the side chain on proton transport were investigated. The results showed that the introduction of the imidazole and bipyridine group can effectively increase the phosphoric acid absorption of the membrane. The proton conductivity of the COPBI-20 membrane under saturated PA doping was as high as 0.105 S cm^−1^ at 160 °C and the maximum output power density reached 836 mW cm^−2^ at 160 °C, which was 2.3 times that of the OPBI membrane and 1.3 times that of the COPBI-0-containing bipyridine group. Notably, even at low ADL, while maintaining the high tensile strength near 22.2 MPa, the proton conductivity of COPBI-20 achieved 0.054 S cm^−1^ at 160 °C, which was 2.3 times that of the OPBI membrane, and the maximum output power density of a single cell reached 540 mW cm^−2^ at 160 °C, which was 1.5 times that of the OPBI membrane. The results suggest that the introduction of a large number of nitrogen-containing sites in the main chain and side chain with low acid doping is an efficient way to avoid the trade-off between conductivity and mechanical strength.

## Figures and Tables

**Figure 1 polymers-14-02621-f001:**
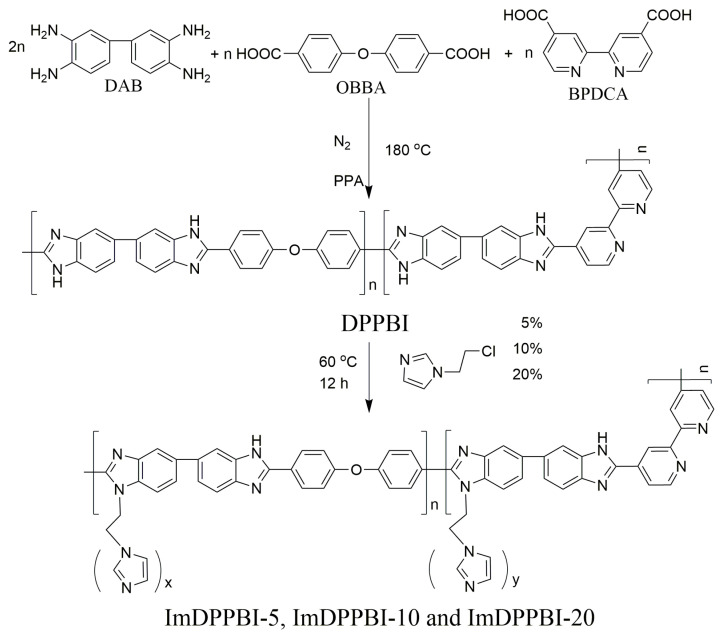
Schematic of PBI synthesis.

**Figure 2 polymers-14-02621-f002:**
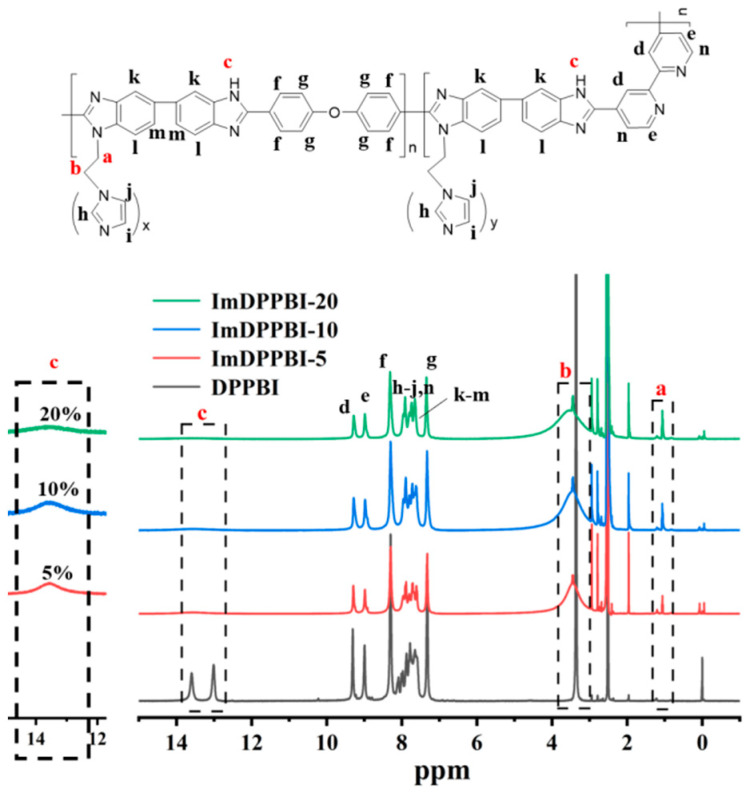
^1^H NMR spectra of the DPPBI and grafted ImDPPBI.

**Figure 3 polymers-14-02621-f003:**
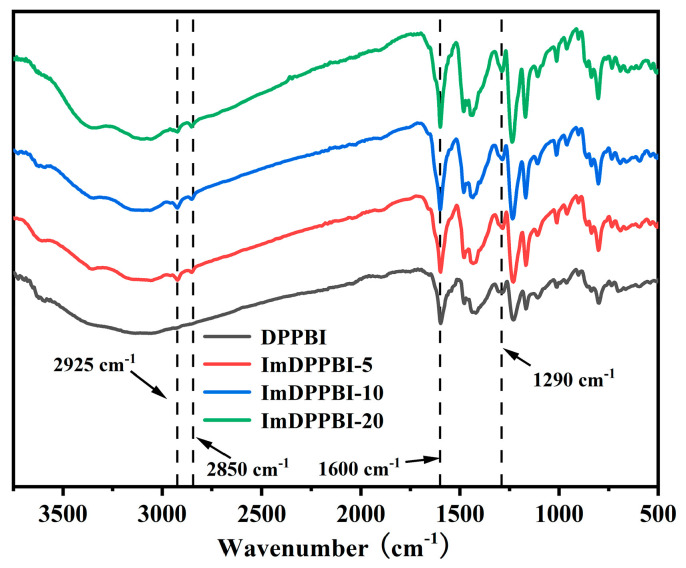
FT–IR spectra of the DPPBI and grafted ImDPPBI.

**Figure 4 polymers-14-02621-f004:**
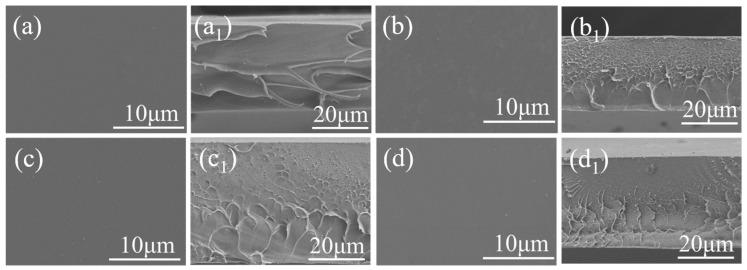
Surface and section morphology of COPBI-0 (**a**,**a_1_**), COPBI-5 (**b**,**b_1_**), COPBI-10 (**c**,**c_1_**), and COPBI-20 (**d**,**d_1_**).

**Figure 5 polymers-14-02621-f005:**
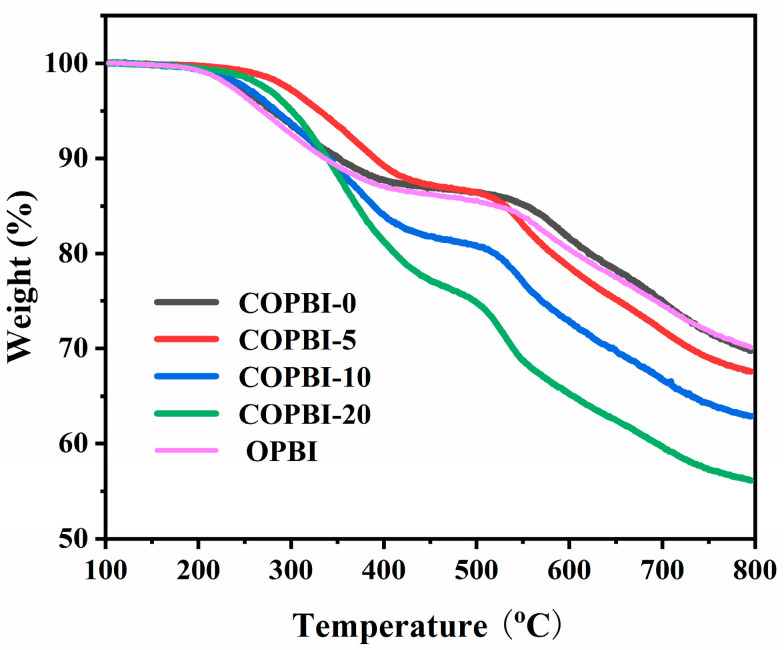
Thermogravimetric curves of OPBI and COPBI membranes.

**Figure 6 polymers-14-02621-f006:**
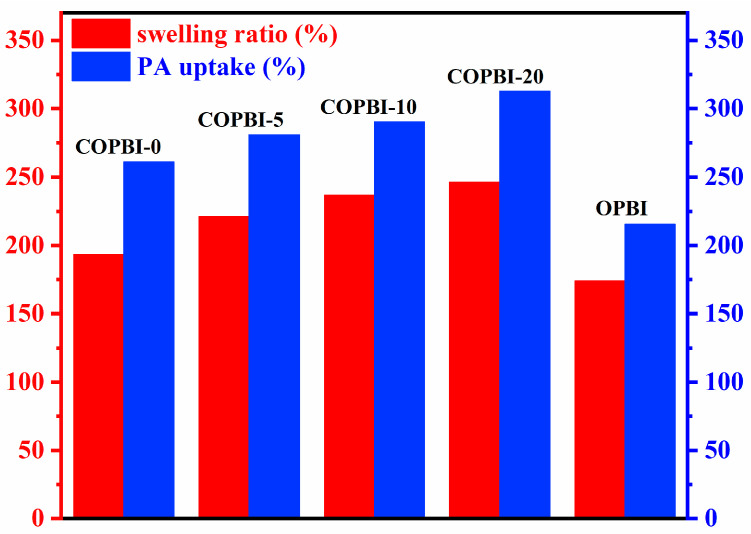
PA uptake and swelling ratio of OPBI and COPBI membranes.

**Figure 7 polymers-14-02621-f007:**
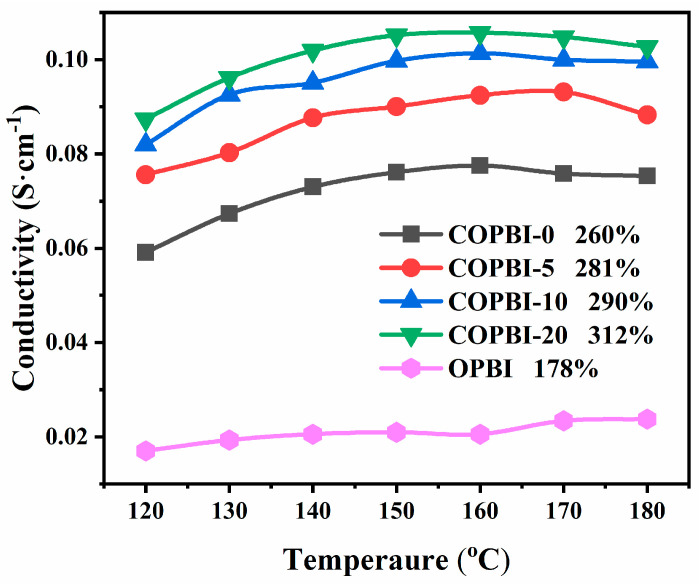
Proton conductivity of OPBI and COPBI membranes at saturated absorption.

**Figure 8 polymers-14-02621-f008:**
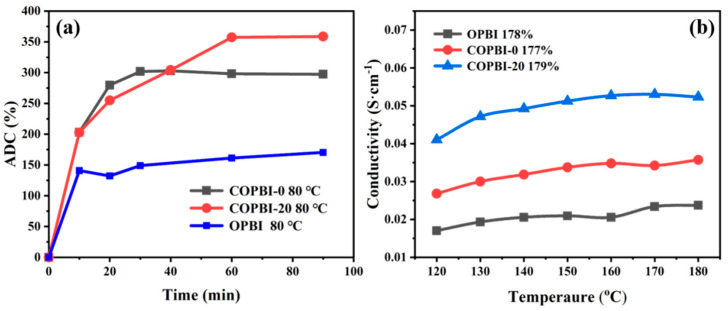
(**a**) Phosphoric acid absorption curve of OPBI, COPBI-0, and COPBI-20 membranes at 80 °C. (**b**) Comparison of proton conductivities of OPBI, COPBI-0, and COPBI-20 under similar ADC.

**Figure 9 polymers-14-02621-f009:**
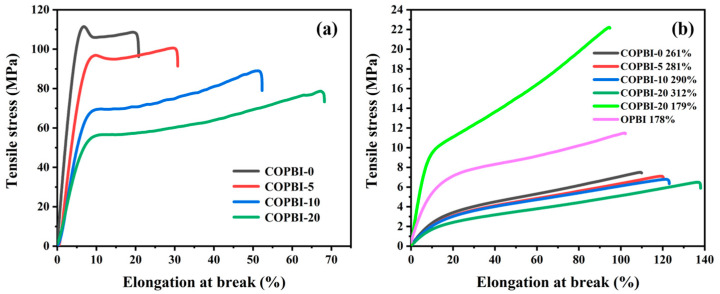
(**a**) Mechanical property curves of OPBI and COPBI membranes. (**b**) Mechanical property curves of OPBI and COPBI membranes with different ADC.

**Figure 10 polymers-14-02621-f010:**
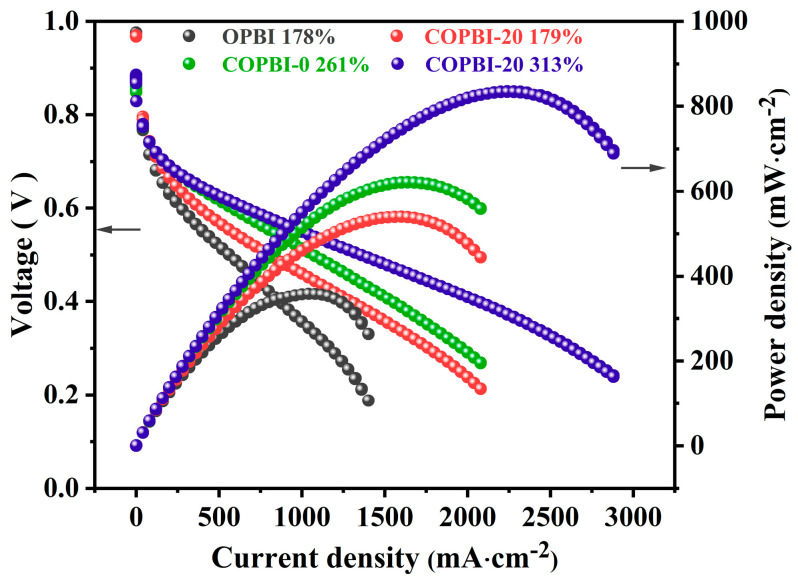
Single cell performance of OPBI and COPBI membranes.

**Table 1 polymers-14-02621-t001:** Mechanical properties, ADC, and swelling rate of all membranes.

Membrane	ADC	Swelling	Tensile Strength (MPa)		Elongation at Break (%)	
	(%)	V (%)	Undoped	Doped	Undoped	Doped
OPBI	178 ± 6	174 ± 9	65.6 ± 0.6	11.5 ± 0.3	26.0 ± 0.6	100.9 ± 1.9
COPBI-0	261 ± 2	193 ± 4	112.9 ± 6.2	8.2 ± 0.7	21.0 ± 7.5	119.1 ± 9.1
COPBI-5	281 ± 6	221 ± 2	102.5 ± 5.1	7.4 ± 0.3	28.2 ± 7.3	128.8 ± 8.5
COPBI-10	290 ± 1	237 ± 2	86.0 ± 3.0	6.9 ± 0.1	43.6 ± 8.7	133.0 ± 9.9
COPBI-20	313 ± 3	246 ± 4	74.7 ± 4.0	6.5 ± 0	59.5 ± 8.8	138.1 ± 0
COPBI-20	179 ± 5	198 ± 5	74.7 ± 4.0	22.2 ± 0.9	59.5 ± 8.8	94.8 ± 8.2

**Table 2 polymers-14-02621-t002:** Comparison of the performance of PA-PBI membranes.

Membrane	ADC	Tensile Strength	Proton Conductivity	Power Density	Single Cell Test Conditions			Reference
	(%)	(MPa)	(S cm^−1^)	(mW cm^−2^)	Test Gas	Pt Catalyst Concentration (mg cm^−2^) Anode/Cathode	Temperature (°C)	
COPBI-0	261 ± 2	8.2 ± 0.7	0.077 (160 °C)	622	H_2_/O_2_	1.0/1.0	160	This work
COPBI-20	313 ± 3	6.5 ± 0	0.105 (160 °C)	836	H_2_/O_2_	1.0/1.0	160	This work
COPBI-20	179 ± 5	22.2 ± 0.9	0.054 (160 °C)	540	H_2_/O_2_	1.0/1.0	160	This work
OPBI	178 ± 6	11.5 ± 0.3	0.023 (170 °C)	358	H_2_/O_2_	1.0/1.0	160	This work
PDA-PBI	237	12.3	0.082 (160 °C)	460	H_2_/O_2_	0.6/0.6	160	[18]
g3-OPBI	252	4.1	0.101 (160 °C)	305	H_2_/O_2_	0.6/0.6	160	[22]
g-PBI-20	381	6.5	0.117 (120 °C)	443	H_2_/O_2_	0.6/0.6	160	[28]

## Data Availability

The data presented in this study are available on request from the corresponding authors.

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
