# Peer review of "Nitrogen Dense Distributions of Imidazole Grafted Dipyridyl Polybenzimidazole for a High Temperature Proton Exchange Membrane"

_polymers, 2022, doi:10.3390/polym14132621_

Round 1
Reviewer 1 Report
Dear Authors
In your work, the introduction of basic groups in the polybenzimidazole (PBI) main chain or side chain was used as an effective way to avoid the trade-off between proton conductivity and mechanical strength for high temperature proton exchange membrane (HT-PEM). In this study, the ethyl imidazole is grafted on the side chain of the PBI containing bipyridine in the main chain, and blended with poly(2,2’-[p-oxydiphenylene]-5,5’-benzimidazole) (OPBI) to obtain a series of PBI composite membranes for HT-PEMs. The effects of the introduction of bipyridine in the main chain and the ethyl imidazole in the side chain on proton transport are investigated. The result suggests that the introduction of the imidazole and bipyridine group can effectively improve the comprehensive properties as HT-PEM. The highest of proton conductivity of the obtained membranes under saturated phosphoric acid (PA) doping achieves as high as 0.105 S cm−1 and the maximum output power density is 836 mW cm−2, which is 2.3 times that of the OPBI membrane. Importantly, even at low acid doping content (~178%), the proton conductivity of the membrane achieves 0.054 S cm−1, which is 2.3 times that of the OPBI membrane, and the maximum output power density of a single cell is 540 mW cm−2, which is 1.5 times that of the OPBI membrane. The results suggest that the introduction of a large number of nitrogen-containing sites in the main chain and side chain is an efficient way to improve the proton conductivity even at low PA doping level.
The manuscript is very interesting for the readers and well designed and presented.
Only a minor revision is required before considering publication.
The title should contain some indication about the developed composite membranes.
Figure 6 caption is a dublicate of Figure 5. Please correct.
Author Response
1. The title should contain some indication about the developed composite membranes.
Thank you very much for your valuable suggestion. The title of the manuscript has been revised as “Nitrogen Dense Distributions of Imidazole Grafted Dipyridyl Polybenzimidazole for High Temperature Proton Exchange Membrane”
2. Figure 6 caption is a dublicate of Figure 5. Please correct.
Sorry for our carelessness. The caption has been corrected as “PA uptake and swelling ratio of OPBI and COPBI membranes”.
Reviewer 2 Report
The manuscript reported the fabrication of PBI composite membrane with low phosphoric acid doping level and high proton conductivity. The introduction of bipyridine and ethyl imidazole into the main and side chain, respectively, can improve the proton conductivity even at low acid doping level. The manuscript is well written and recommended for the publication in the journal after some minor revision.
- Abstract. It is suggested to add the temperature at which the data was obtained.
- Experimental section. Page 3 line 89: "...polyphosphoric acid (PPA) was added to a 100 mL 89 three-necked flask, mechanical stirred under nitrogen atmosphere..." Was it added into the empty flask? Then why was it stirred? If not, please specify the other components in the flask.
- Page 4, NMR results. It is suggested to add the chemical formulas of the components with the numbered hydrogens, so it will be more clear to understand and easier to follow.
- Page 6 line 169: "...COPBI-5 > COPBI-20 > COPBI-20 > COPBI-0..." Please check the details.
- Conclusions. Again, please specify the temperature for the data provided.
- Supplementary. Page 3. Tensile strength and elongation at break method is repeated twice.
Author Response
1. Abstract. It is suggested to add the temperature at which the data was obtained.
Response: Thank you for your valuable suggestion. The test temperatures of proton conductivity and single cell performance have been added in the abstract.
2. Experimental section. Page 3 line 89: "...polyphosphoric acid (PPA) was added to a 100 mL 89 three-necked flask, mechanical stirred under nitrogen atmosphere..." Was it added into the empty flask? Then why was it stirred? If not, please specify the other components in the flask.
Response: In the first step, only polyphosphoric acid (PPA) without any other components was added in the flask, mechanical stirred under nitrogen atmosphere for 1 h until the PPA was homogeneous, transparent and free of bubbles. The reason for this step is that the PPA is viscous and the air may be encased in the solution, the stirring under nitrogen atmosphere can remove the oxygen from the solution to ensure the polymerization in the next step without the influence of oxygen.
3. Page 4, NMR results. It is suggested to add the chemical formulas of the components with the numbered hydrogens, so it will be more clear to understand and easier to follow.
Response: Thanks a lot for your valuable suggestion. The chemical formulas have been added in Figure 2.
4. Page 6 line 169: "...COPBI-5 > COPBI-20 > COPBI-20 > COPBI-0..." Please check the details.
Response: Sorry for our carelessness. The sentence has been corrected as “COPBI-5 > COPBI-20 > COPBI-10 > COPBI-0 > OPBI”.
5. Conclusions. Again, please specify the temperature for the data provided.
Response: The temperatures of the tests have been added in conclusions section.
6. Supplementary. Page 3. Tensile strength and elongation at break method is repeated twice.
Response: Sorry for the mistake caused by our carelessness. The repeat sentence has been deleted.
Reviewer 3 Report
In this paper, the authors prepared proton exchange membranes (PEMs) in which dypyridyl and ethyl imidazole moieties were introduced into polybenzimidazole (PBI) and reported the results of application to high-temperature PEMFC. It showed a significant improvement in performance compared to the existing PBI membranes, and it is thought that the paper was well organized. However, in order to be published in "polymers", it must be revised in consideration of the following points.
1. (L57) trizole -> triazole
2. (L61) et al (missing period)
3. (L169-170) COPBI-20 was duplicated
4. In the results of Figures 5-9 and Table 1, the data of no dipyridyl group (i.e. OPBI) should also be compared. In this way, the effect of the introduction of the dipyridyl group on the membrane performance can be clearly confirmed.
5. In Figures 7 & 8, the graph should be indicated by a solid line.
6. In this study, the authors tried to avoid the trade-off between proton conductivity and mechanical strength of the membranes, but the results in Figures 7 & 9 show that the trade-off relationship strengthens as the degrees of grafting increases. A detailed explanation is required for this.
7. It is necessary to evaluate the long-term stability of the PEMs developed in this paper.
Author Response
- (L57) trizole -> triazole
Response: Sorry for the mistake caused by our carelessness. The trizole has been revised as triazole.
- (L61) et al (missing period)
Response: Sorry for our carelessness. The period has been added after et al.
- (L169-170) COPBI-20 was duplicated
Response: The sentence has been corrected as “COPBI-5 > COPBI-20 > COPBI-10 > COPBI-0”.
- In the results of Figures 5-9 and Table 1, the data of no dipyridyl group (i.e. OPBI) should also be compared. In this way, the effect of the introduction of the dipyridyl group on the membrane performance can be clearly confirmed.
Response: Thank you very much for your valuable suggestion. The data of OPBI have been added in Figure 5-9 and Table 1. The corresponding discussion and comparison were added in the revised manuscript. We also compared the comprehensive properties of the PA doped COPBI-0 and COPBI-20 including the rensile strength, ADC, proton conductivity and the highest power density were compared with those of commercial available OPBI and other published literatures related to pyridine containing or grafting PBI membranes as shown in Table 2.
- In Figures 7 & 8, the graph should be indicated by a solid line.
Response: The Figure 7 and 8 have been indicated by solid lines as seen in the revised manuscript.
- In this study, the authors tried to avoid the trade-off between proton conductivity and mechanical strength of the membranes, but the results in Figures 7 & 9 show that the trade-off relationship strengthens as the degrees of grafting increases. A detailed explanation is required for this.
Response: Sorry for our unprecise explanation. The introduction of basic groups in the polybenzimidazole (PBI) main chain or side chain with low phosphoric acid (PA) doping is an effective way to avoid the trade-off between proton conductivity and mechanical strength for high temperature proton exchange membrane. The mechanical properties of COPBI membranes with low PA doping have been supplemented in the revised manuscript. With the increase of grafting degree, the conductivity of the membrane gradually increased, and the highest proton conductivity increased from 0.077 S cm−1 of COPBI-0 to 0.105 S cm−1 of COPBI-20, which indicates the increase of basic imidazole group branched on the side chain can increase the density of the hydrogen bond network in the membrane after absorbing PA, thereby increasing the proton transport paths and improving the proton transport efficiency. However, the density of basic N sites also trend to absorb more PA which will decrease the mechanical strength of the membranes. Therefore, low PA doped COPBI membranes were prepared and the mechanical properties were investigated. Interestingly, the COPBI membranes with low ADL exhibited higher tensile strength. The tensile strength of the COPBI-20 with 179%ADC membrane was 22.2 MPa, which was nearly 2 times that of OPBI.
- It is necessary to evaluate the long-term stability of the PEMs developed in this paper.
Response: Thank you for your valuable suggestion. The single cell test instrument in our institute is fully booked in next 3 months. Once the instrument is available, we will run the long-term stability test as soon as possible. We appreciate your understanding.
Round 2
Reviewer 3 Report
The authors have carefully revised the manuscript according to the referees’ comments. In my opinion, this manuscript could be accepted for publication in "Polymers".